# Regular Exercise Training Induces More Changes on Intestinal Glucose Uptake from Blood and Microbiota Composition in Leaner Compared to Heavier Individuals in Monozygotic Twins Discordant for BMI

**DOI:** 10.3390/nu16203554

**Published:** 2024-10-20

**Authors:** Martin S. Lietzén, Maria Angela Guzzardi, Ronja Ojala, Jaakko Hentilä, Marja A. Heiskanen, Sanna M. Honkala, Riikka Lautamäki, Eliisa Löyttyniemi, Anna K. Kirjavainen, Johan Rajander, Tarja Malm, Leo Lahti, Juha O. Rinne, Kirsi H. Pietiläinen, Patricia Iozzo, Jarna C. Hannukainen

**Affiliations:** 1Turku PET Centre, University of Turku, 20521 Turku, Finlandjhannukainen@gmail.com (J.C.H.); 2Institute of Clinical Physiology, National Research Council, 56124 Pisa, Italy; 3Research Centre of Applied and Preventive Cardiovascular Medicine, University of Turku, 20521 Turku, Finland; 4Centre for Population Health Research, University of Turku and Turku University Hospital, 20520 Turku, Finland; 5Heart Centre, Turku University Hospital, 20521 Turku, Finland; 6Department of Biostatistics, University of Turku, 20520 Turku, Finland; 7Turku PET Centre, Radiopharmaceutical Chemistry Laboratory, University of Turku, 20521 Turku, Finland; 8Turku PET Centre, Accelerator Laboratory, Åbo Akademi University, 20500 Turku, Finland; 9A.I. Virtanen Institute for Molecular Sciences, University of Eastern Finland, 70211 Kuopio, Finland; 10Department of Computing, University of Turku, 20521 Turku, Finland; 11Turku PET Centre, Turku University Hospital, 20520 Turku, Finland; 12Obesity Research Unit, Research Program for Clinical and Molecular Metabolism, Faculty of Medicine, University of Helsinki, 00014 Helsinki, Finland; 13Abdominal Center, Obesity Center, Endocrinology, University of Helsinki and Helsinki University Central Hospital, 00014 Helsinki, Finland

**Keywords:** physical exercise, gut microbiota, insulin sensitivity, intestine

## Abstract

Background/Objectives: Obesity impairs intestinal glucose uptake (GU) (intestinal uptake of circulating glucose from blood) and alters gut microbiome. Exercise improves intestinal insulin-stimulated GU and alters microbiome. Genetics influence the risk of obesity and gut microbiome. However, the role of genetics on the effects of exercise on intestinal GU and microbiome is unclear. Methods: Twelve monozygotic twin pairs discordant for BMI (age 40.4 ± 4.5 years, BMI heavier 36.7 ± 6.0, leaner 29.1 ± 5.7, 8 female pairs) performed a six-month-long training intervention. Small intestine and colonic insulin-stimulated GU was studied using [^18^F]FDG-PET and microbiota from fecal samples with 16s rRNA. Results: Ten pairs completed the intervention. At baseline, heavier twins had lower small intestine and colonic GU (*p* < 0.05). Response to exercise differed between twins (*p* = 0.05), with leaner twins increasing colonic GU. Alpha and beta diversity did not differ at baseline. During the intervention, beta diversity changed significantly, most prominently at the mid-point (*p* < 0.01). Beta diversity changes were only significant in the leaner twins when the twin groups were analyzed separately. Exercise was associated with changes at the phylum level, mainly at the mid-point (pFDR < 0.05); at the genus level, several microbes increased, such as *Lactobacillus* and *Sellimonas* (pFDR < 0.05). In type 1 analyses, many genera changes were associated with exercise, and fewer, such as *Lactobacillus*, were also associated with dietary sugar consumption (*p* < 0.05). Conclusions: Obesity impairs insulin-stimulated intestinal GU independent of genetics. Though both twin groups exhibited some microbiota changes, most changes in insulin-stimulated colon GU and microbiota were significant in the leaner twins.

## 1. Introduction

The intestine is a key organ affecting whole-body energy homeostasis through nutrient absorption and gut hormone secretion [1]. Obesity and type 2 diabetes have been shown to deteriorate insulin-stimulated intestinal glucose uptake (intestinal uptake of circulating glucose from blood) [2,3]. Intestinal microbiota has a role in human health by assisting with digestion and host immune function [4,5,6] and its composition has been shown to differ between healthy, obese, and individuals with insulin resistance [7,8].

The benefits of regular exercise training on overall health are widely known. Studies on the effects of exercise on intestinal substrate uptake are limited but short-term moderate-intensity continuous training and weight loss have been shown to improve intestinal insulin-stimulated glucose uptake (GU) [3,9]. Exercise has been shown to alter gut microbiome composition [10]. Exercise has also been shown to affect markers of healthy gut microbiome, such as the microbiome diversity, which has been greater in athletes compared to sedentary controls in some [11,12] but not all studies [13]. Cardiorespiratory fitness and endurance training have been shown to influence gut microbial diversity in humans [10,14,15], but there are also studies in which the relationship was mediated by body composition [16,17]. The role of strength training in microbiome composition is not as clear as there are studies where the gut microbiome of bodybuilders did not differ significantly from a more sedentary control group [13,18].

Genetics play a significant role in the susceptibility towards the risk of obesity and type 2 diabetes [19] and influence the individual response to exercise [20]. Genetics define the chemical and physical environment inhabited by microbiome with nutrient availability and the activity of the immune system [21]. Microbiome genome-wide association studies have also identified numerous associations between human genetic variants and gut microbiome composition [22,23]. Genetics also possibly affect the risk of some diseases by microbiota composition [24]. Some genetic variants affect the immune system, causing differences in sensing and responding to micro-organisms, which may cause disproportionate immune responses implicated in the pathogenesis of inflammatory bowel disease [24]. Gut microbiome composition in both monozygotic (MZ) and dizygotic twins show similarities compared to unrelated individuals [25,26], indicating that there is a heritability component in gut microbiome [27]. However, there are also studies where environmental factors, such as a shared household, are highlighted as the key determinants for microbiome similarities [28].

MZ twins enable the effects of exercise to be studied with minimal confounding effects from genetics. MZ twins also often share the same environmental factors until adulthood. Therefore, MZ twins provide a good effective model to evaluate the independent response to acquired obesity and to a lifestyle intervention.

In this study, we examined the effects of obesity on insulin-stimulated intestinal GU from blood (small intestine and colon) and microbiota composition in monozygotic twins discordant for body mass index (BMI). We also studied whether these changes can be altered with a long-term regular exercise training intervention. We used non-invasive imaging of 2-deoxy-2-[^18^F]fluoro-D-glucose ([^18^F]FDG) by positron emission tomography (PET) during euglycemic hyperinsulinemic clamp and computed tomography (CT) and collected stool samples for microbiota analysis. We hypothesized that the heavier twins would have a decreased insulin-stimulated intestinal GU and microbiota diversity compared to their leaner co-twins and that exercise would induce improvements in both twins, especially in the heavier twins.

## 2. Materials and Methods

### 2.1. Study Design

This study is part of a larger study entitled Systemic cross-talk between brain, gut, and peripheral tissues in glucose homeostasis: effects of exercise training (CROSSYS) (NCT03730610) [29], an exercise training intervention conducted at Turku PET Centre (Turku, Finland). The Ethical Committee of the Hospital district of South-Western Finland (100/1801/2018/438§) approved the study protocol, informed consent, and patient information of CROSSYS. Approval date: 23 November 2018. Good Clinical Practices and the Declaration of Helsinki were followed while conducting this study. An informed consent was signed by all participants before participating in this study. A detailed description of the methods and design of this study has been published [29,30]. MZ twin pairs discordant for BMI were studied to minimize the confounding effects of genetics. Clinical studies were carried out between 1/2019 and 10/2021. The datasets generated during the current study are available from the corresponding author upon reasonable request.

### 2.2. Participants

Three population-based longitudinal twin studies were used to recruit MZ twins discordant for BMI [29]. All twins were identified through the Finnish central population registry. A total of 55 MZ pairs discordant for BMI and/or insulin resistance had previously participated in the intensive metabolic study arm at the University of Helsinki, where their monozygosity was confirmed by the genotyping of ten informative genetic markers [31]. Of the total of 55 MZ pairs, 12 pairs were available and willing to participate and met the inclusion criteria [29], and 10 pairs completed this study [30]. Five of the leaner co-twins had impaired fasting glucose (IFG), and two had impaired glucose tolerance (IGT), as defined by the American Diabetes Association guidelines [32]. Seven of the heavier co-twins had IFG, two had IGT, and two were treated for hypertension. No participants had diabetes or were treated for hyperlipidemia. All female participants (eight pairs) were premenopausal. One of the heavier twins finished an antibiotic treatment consisting of trimethoprim and sulfadiazine 13 days before giving the post feces sample.

### 2.3. Training Intervention

All study participants performed the training intervention with a mean duration of 27 (SD 2) weeks, exercising four times a week in their place of residence with one training session supervised by a local personal trainer and three unsupervised home-based exercise training sessions, as detailed previously [29]. The training consisted of the following sessions: two endurance, one resistance training, and one high-intensity interval training per week. The training intensity was increased progressively, and individual loads were determined with a personal trainer. Training intensity, duration, and adherence were monitored using a heart rate monitor (Polar A370, Polar, Finland).

### 2.4. Euglycemic Hyperinsulinemic Clamp and FDG-PET Study

Detailed FDG-PET/CT-scan protocol has been previously described [29]. In brief, intestinal insulin-stimulated GU was studied by FDG-PET during an euglycemic hyperinsulinemic clamp [33]. Participants fasted overnight (at least 10 h) and avoided excess physical activity for 48 h before the FDG-PET study. They were positioned supine into the PET scanner (Discovery MI (DMI), GE Healthcare, Chicago, IL, USA) and instructed to avoid excess muscle contractions. The euglycemic-hyperinsulinemic clamp was performed, as originally described by Defronzo et al. [34]. Once steady-state in blood glucose was achieved, [^18^F]FDG was injected [29], and the abdominal area PET scan started 56 (SD 3.2) minutes post-injection. Blood samples were taken regularly to monitor blood glucose, insulin, free fatty acid levels, and plasma radioactivity (input function) [29]. Small intestine and colon PET images were analyzed using Carimas software 2.10.3.0 (http://turkupetcentre.fi accessed on 23 September 2019). CT images were used as an anatomical reference alongside PET images. Small intestine activity values were averaged over four different regions of interest (ROI). One cylinder and one tube-shaped three-dimensional ROI was placed into the upper left quadrant of the abdomen, and one cylinder and one tube-shaped ROI was placed similarly in the lower-left quadrant of the abdomen. The left quadrant of the abdomen was chosen to avoid spillover effects from organs with high GU, such as the liver. The ROI were manually placed following the CT anatomical reference images to the target organs and shaped to follow the anatomical landmarks as precisely as possible. The PET images were used to check that there were no clear high-intensity radioactive spots that would indicate either a different organ or tissue. With ROI shape-modifying tools in Carimas software, it was also made sure that the distance to organs with high GU, such as the liver, was maximized. The final GU was calculated as an average of at least two ROI from different parts of the same organ to minimize the potential spillover from surrounding organs. Colon activity values were averaged over one cylinder-shaped and one tube-shaped ROI (cylinder in the right side and tube in the left side of the transversum colon). A fractional uptake rate was used to determine GU.

### 2.5. Magnetic Resonance Imaging

Visceral fat mass was defined using magnetic resonance imaging (MRI), approximately four hours after lunch. The Siemens magnetom Skyra fit 3 T MRI system (Siemens Healthcare, Erlangen, Germany) was used for imaging, as described previously [29]. The visceral fat was segmented from the fat fraction maps by drawing 2-dimensional ROI on every 5–9 slices and creating a 3-dimensional volume of interest (VOI) from the slices with Carimas software using the interpolation feature. Then, all voxels with an intensity value below 0.5 (i.e., fat fraction over 50%) were excluded, and the remaining VOI was regarded as visceral adipose tissue.

### 2.6. Body Composition and Peak Aerobic Exercise Capacity Test

A bioimpedance analysis machine (Inbody 720; Biospace Co., Seoul, Republic of Korea) was used to measure body composition. Peak exercise capacity (VO_2peak_) was measured using a stationary bicycle ergometer test (Ergoline 800 s, VIASYS Healthcare, Hochberg, Germany) until volitional exhaustion. More in-depth details concerning the test have been previously described [29].

### 2.7. Dietary Assessment

Participants were asked not to change their habitual dietary intake or physical activity outside of the intervention during the intervention. The participants completed a food diary for 3–4 days at baseline, mid, and post-intervention. Participants were instructed to complete the diary close to the intervention time points but could choose the exact days themselves. Food diaries were instructed to be completed before giving the feces sample. Depending on the chosen time point, there were both working days and non-working days included in the diary. Portion size was estimated by the participant using applicable measurement units such as deciliters or grams. Macro- and micro-nutrients were calculated based on the completed diaries using the online food diary calculator provided by Fineli from the Finnish Institute for Health and Welfare.

### 2.8. Analysis of Fecal Microbiota

The participants gave feces samples at baseline, mid, and post-intervention. Participants were instructed to bring a sample collected at the same time and day when they came for pre and post-visits on-site. Samples were allowed to be at room temperature for a maximum of four hours.

Mid-samples were instructed to be collected at home in a container given by the researchers. The mid samples were stored at the participants’ homes in a regular freezer (−20 °C) and brought during the post-visit. Participants were instructed to store the mid-sample in a cooler bag during transportation to the study site.

All the samples were collected to a sterile feces tube with a spoon and a screw cap (Sarstedt Group with its head office in Nümbrecht, Germany).

After receiving the samples, they were stored in a freezer (−70 °C). DNA libraries were performed with the amplification of the Specific 16S ribosomal RNA (rRNA) gene region (V3–V4) following Illumina protocols using PE250 Illumina and SILVA taxonomy 138 database, QIIME 2.0. A more detailed description of the processing of the fecal matter and taxonomic assignment has been previously published [29].

### 2.9. Statistical Analysis and Modelling

Normality assumption was checked from studentized residuals, and logarithmic transformations were performed to fulfill the normal distribution when needed. A linear mixed model for repeated time points using compound symmetry covariance structure was used to analyze the data. The model consisted of two within-factors: time (before and after intervention) and group (heavier and leaner co-twin), and their interaction term (time × group). For differences between groups at baseline, the same model was used but only using the pre-intervention results. If, after the intervention, there was a significant time × group result, the same model was used to determine the within-group time effect to see if one of the groups had a significant time effect or if the groups responded differently to the intervention. Missing data points from participants were included in the statistical analysis by using the restricted maximum likelihood estimation within the linear mixed models. Therefore, model-based means and 95% confidence intervals are reported. The same statistical analyses were performed for the gut microbiota alpha diversity indices, which were carried out at the amplicon sequencing variant (ASV) level. All statistical tests were performed as two-sided, and *p*-values less than 0.05 were considered statistically significant. The SAS System, version 9.4 for Windows (SAS Institute, Cary, NC, USA), was used for the analyses.

Gut microbiota underwent total-sum scaling (TSS) normalization. Then, beta diversity was computed based on the Bray–Curtis index, Jensen–Shannon Divergence, and Jaccard Index and visualized on a Principal Coordinate Analysis (PCoA) plot. Group differences in the relative taxon abundances were analyzed at phylum, genus, and ASV taxonomic levels by multivariate analysis. False discovery rate (FDR) correction was applied to limit the number of false positives. FDR-corrected *p*-values ≤ 0.05 were considered statistically significant. Analyses were conducted on the free online software MicrobiomeAnalyst, version 2.0 [35].

The association of exercise and nutrition to microbiota was analyzed using an additional calculation method (type 1 method), where effects are tested sequentially, fitting one additional effect at each step. With this method, we can see whether the following explanatory variable is significant when the model includes all other explanatory variables, i.e., all variations explained by earlier factors have already been taken into account. These analyses were carried out using a JMP^®^ Pro 16.0.0 for Windows (SAS Institute, Cary, NC, USA).

## 3. Results

In the baseline results, the heavier twins had both a lower VO_2peak_ and a whole-body insulin-stimulated glucose uptake (M-value) (Table 1) compared to the leaner co-twins. The exercise intervention improved the VO_2peak_ and M-value in both groups (time *p* < 0.05 in both variables). Exercise had no effect on whole-body mass or fat percentage, but it tended to decrease visceral fat mass in the whole population (time *p* = 0.07).

The heavier twins had lower small intestine GU at baseline, but no training-induced changes were seen in either group after the intervention (Figure 1).

Colon GU was lower in the heavier compared to the leaner twins at baseline (*p* = 0.005) and the training response differed between groups after the intervention (time × group *p* = 0.05). When the groups were analyzed separately, colon GU only increased in the leaner twins after the intervention (*p* = 0.023) (Figure 2). At baseline, BMI had a significant inverse correlation with the small intestine GU and colon GU (*p* < 0.01, R < −0.68 in both).

Gut microbiota composition was analyzed at baseline, at 16.6 (SD 2.7) weeks after the start of the intervention (mid-point), and at the end of the intervention. At baseline, there were no differences in alpha diversity between the twins. After the intervention, only Pielou’s evenness (measurement of how similar or dissimilar the relative abundances of the species in a set are) decreased significantly (*p* = 0.02) in the whole population (Table 2).

No difference was observed in the beta-diversity between the twins at baseline, but beta-diversity changed significantly after the training intervention in the whole population. Indeed, PCoA of inter-sample variation based on the Bray–Curtis index, Jensen–Shannon Divergence, and Jaccard index showed significant segregation at all the analyzed taxonomic levels (*p* = 0.001 for all distance metrics at phylum level, at genus and ASV levels, *p*-value below 0.05 for Jaccard Index only) between the three time points, with the main change observed at the mid-point, which was significantly different from both baseline and post-intervention time points (*p* < 0.05 in all distance metrics, at phylum, genus, and ASV levels) (Figure 3).

When the twin groups were analyzed separately, beta diversity over time was significantly different in the leaner twins (*p* < 0.05 for all dissimilarity indexes) but not in the heavier twins (*p* > 0.05 for all), suggesting that the training-induced changes in relative taxon abundances were mostly driven by the lean twins (Figure 4).

Then, we investigated the taxonomic composition of the heavier and leaner twins and their change over time. In accordance with the diversity results, the relative taxon abundances were similar between twins at baseline. In the whole population, the intervention led, at the mid-point, to a reduction in Bacteroidota and Proteobacteria phyla and an increment in Firmicutes phylum (pFDR < 0.05). However, at the end of the intervention, all three phyla (Bacteroidota, Proteobacteria, and Firmicutes) were back to baseline levels. At the end of the intervention, only Campylobacterota, a bacterial phylum of low relative abundance, showed a significant increase compared to baseline. At the genus level, the intervention led to the increment of *Megamonas*, *Helicobacter*, *Sellimonas*, *Lactobacillus*, *CHKCL001*, *Cutibacterium*, *Xanthomonas*, *Enorma,* and *Staphylococcus* (pFDR < 0.05 for all) (Figure 5).

The analysis of taxonomic composition in the two twin groups showed that the transient (at mid-point only) increment of Firmicutes and reduction of Bacteroidota and Proteobacteria (trend, pFDR = 0.09) phyla were similar between twins but reached statistical significance only in the leaner group, with the exception of the reduction of Proteobacteria (Figure 6). Similarly, the increment in Campylobacterota at the end of the intervention was statistically significant in the leaner but not the heavier twins.

At the genus level, *Megamonas* abundance was significantly increased in the whole population, whereas the increment of *Lactobacillus* genera, although showing a similar trend in both the heavier and the leaner twins, only reached a statistical significance in the leaner twins (Figure 6).

### Dietary Assessment

No difference in daily average energy intake was observed between the leaner and heavier twin groups at different intervention time points (Table 2). At baseline, the daily average of sugar intake was similar between the twins. After the intervention, both twins tended to increase their intake of sugar (time *p* = 0.068), but the leaner co-twins demonstrated a greater increase in sugar consumption (group × time *p* = 0.043). Fructose intake was similar between the groups at baseline and the intake increased in both groups after the intervention (time *p* = 0.051). Sucrose intake was similar at baseline but after the intervention, the heavier twins had a decrease in average daily intake, whereas their leaner co-twins had an increase (group × time *p* = 0.046) (Table 2).

Dietary fiber intake was higher in the heavier twins at baseline (*p* = 0.053), and both twin groups increased their dietary fiber intake (*p* = 0.054) after the exercise intervention. Heavier twins had a higher intake of insoluble fiber and soluble polysaccharides at baseline (*p* = 0.034 and *p* = 0.030, respectively). After the intervention, there was a tendency towards an increase in insoluble fiber consumption (*p* = 0.09) (Table 2).

In the type 1 analysis (Appendix A), exercise was associated strongly with *Megamonas*, *Helicobacter*, *Sellimonas*, *Enorma*, and *Rikennella* (*p* < 0.031 for all), and no significant association with nutrition was found. *Lactobacillus* was affected by exercise, sugar, glucose, fructose, and sucrose (*p* < 0.025 for all), and the association with exercise was stronger than nutrition (F-ratio between 19.0 and 28.4). Changes in *CHKCL001* were significantly associated with exercise, glucose, and fructose consumption similarly (*p* < 0.016 for all). Exercise and sugar were associated with *Xanthomonas* (*p* < 0.046 in both), but the association with exercise was greater (F ratio 30.1 vs. 4.2). Glucose and sucrose tended to associate with *Xanthomonas* (*p* = 0.08 and *p* = 0.07, respectively). Exercise, glucose, and fructose all individually affected *Cutibacterium* and *Staphylococcus* in a similar way (*p* < 0.035 for all). Dietary fiber had a weak but significant association with *Cutibacterium* (*p* = 0.007, F-ratio 0.74) and tended to affect *Staphylococcus* (*p* = 0.053).

## 4. Discussion

This study shows that whole-body insulin sensitivity and tissue-specific intestinal insulin sensitivity are impaired in obesity. At baseline, the small intestine and colonic GU from blood was higher in the leaner compared to the heavier twins, while no group difference was observed in microbiota Alpha and Beta diversity. Long-term regular exercise training, without weight loss, improved colon GU and had a significant impact on microbiota composition; however, interestingly, the observed changes in colon GU and microbiota composition were mostly seen in the leaner twins.

### 4.1. Intestinal Glucose Metabolism

Our baseline results align with previous studies showing impaired intestinal GU in obesity, as the heavier twins had lower intestinal GU in both the small intestine and the colon compared to their leaner co-twins [3,36]. In addition, there was a strong inverse correlation between BMI and intestinal GU, confirming that intestinal insulin sensitivity is closely related to obesity. Interestingly, contrary to our hypothesis, the colonic GU improved only in the leaner twins, suggesting that obesity leads to impaired metabolic flexibility in the intestine. It is also important to note that while colonic GU only improved significantly in the leaner twins, both groups showed similar improvements in whole-body insulin sensitivity (M-value).

Previously it has been shown that insulin-stimulated GU in the small intestine is significantly impaired in morbid obesity and that GU is improved after bariatric surgery, leading to weight loss [3]. In the present study, no change was observed in the whole-body mass or fat percentage and the visceral fat mass only tended to decrease, suggesting that the observed improvement in colonic GU was independent of weight loss.

In the present study, training improved GU in the colon but not in the small intestine. Previously, short-term exercise has been shown to improve colonic GU, while the small intestine GU remained unchanged [9]. Moreover, similar findings have been observed after bariatric surgery [3]. It has been suggested that the discrepancy in GU in different parts of the intestine might be due to the differences in the location of GLUT2 receptor prevalence in the enterocytes, as GLUT2 has been observed in the basolateral membrane of an enterocyte in the small intestine (jejunum) [37]; instead, in the colon, GLUT2 is only present in short epithelial portions, involving a limited number of cells [38]. In addition, different digestive tasks between the small and large intestines may play a role in how exercise training strains these mechanisms.

The intestine has been shown to be an insulin-sensitive organ [3,39]. Honka and colleagues showed that insulin was able to increase GU by 2.5- and 2.9-fold in the duodenum and jejunum, respectively, when studied in healthy humans during fasting and insulin stimulation. They further showed in pigs that ^18^F-derived intestinal radioactivity was located in the mucosal layer during fasting and hyperinsulinaemic euglycemia [3,39]. Insulin has been shown to affect GLUT2 receptor location in the enterocyte, as insulin administration leads to internalization of GULT2, resulting in an increase in intestinal GU [40]. Insulin sensitizer medication, such as metformin, has been shown to increase intestinal basolateral GU [41], suggesting that insulin plays a major role in GU from blood to enterocyte. While GLUT2 has a major role in the unidirectional delivery of glucose across the intestine together with SGLT1, it seems to contribute to glucose uptake into the enterocytes, especially during hyperglycemia [37,42]. However, more studies on the mechanisms behind insulin-mediated glucose uptake from blood to enterocytes are needed.

Interestingly, the small intestine GU or colonic GU did not correlate with microbiota at baseline. The relationship between intestinal glucose uptake and microbiota is unclear, but there are preliminary studies showing an association between certain bacterial genera and intestinal glucose uptake measured by FDG-PET in the fasted state [43]. Our results indicate that intestinal GU (glucose uptake from the circulation) is affected more directly by obesity and exercise. In pigs, fecal microbiota has been shown to differ significantly from intestinal microbiota collected from various parts of the gastrointestinal tract [44]. In this study, the microbiota data were derived from fecal samples. Therefore, it is also possible that microbiota collected directly from the small intestine or colon may correlate with intestinal GU. However, due to the methods used in this study, this remains unknown and would be a potential topic for future research.

Our data, combined with previous studies, indicate that small intestine insulin sensitivity is closely related to obesity and that colon GU is more sensitive to training-induced adaptations compared to the small intestine. As there were no significant changes in body composition, the results of our study highlight the beneficial properties of regular long-term exercise to colon GU in leaner twins independent of genetics.

### 4.2. Microbiota Composition

Previously, gut microbiome has been shown to be influenced by host genetics [27,45] and body composition [16,17]. However, there are also studies that attribute environmental factors such as a shared household, diet, and aging as the main components shaping human gut microbiota [28,46]. In this study, there was no difference in alpha and beta diversity between the twin groups at baseline. After the intervention, Pielou´s evenness decreased significantly, and beta diversity had a transient change visible in pre vs. mid and mid vs. post comparisons in the leaner twins. The mid-point change in beta diversity itself can be mainly attributed to the distinct changes in three highly abundant phyla: Firmicutes, Bacteroidota, and Proteobacteria. Proteobacteria and Bacteroidota decreased, whereas Firmicutes increased from pre to mid in the whole study population. Firmicutes have been thought to have a role in obesity as obese people tend to have proportionally more Firmicutes to Bacteroidota than lean people, and a low-calorie diet and weight loss decreases Firmicutes and increases Bacteroidota [47]. Diet has also been shown to affect host bacterial composition, as people eating higher fat and protein diets have more Firmicutes than those with a diet richer in fiber and vegetables [48,49]. In fact, the diet itself, even without changes in body weight, affects the bacterial composition of the host; mice fed with a high-fat diet had an increase in Firmicutes with and without concomitant obesity [50]. Based on these results, it has been proposed that Firmicutes are more effective in extracting energy from food than Bacteroidota [51].

Proteobacteria is a part of the normal human gut microbial community, but the relative abundance varies between healthy, sick, lean, and obese individuals, and increases in the relative abundance of Proteobacteria have been shown in obesity and metabolic disorders [52,53]. Exercise has been shown to affect gut microbiota composition via a decrease in Proteobacteria in obese children and women [54,55]. In our study, there was a distinct decrease in Proteobacteria from pre to mid and a return to baseline from mid to post time points in the whole group. There were no changes in body weight or exercise duration from the first half to the latter half of the intervention, but the average heart rate during the second endurance training workout dropped 3.6 beats per minute in the second half of the intervention, probably due to improved aerobic fitness (Appendix A). As the change in Proteobacteria was transient at the mid-point, the slight decrease in endurance training average heart rate likely did not affect the result. Currently, there is no clear explanation for the decrease in the Proteobacteria in the whole group from pre to mid and whether this can be due to a change in the lifestyle at the beginning of the intervention and a later stabilization of microbiota composition due to long-term exercise training.

Even though both twin groups showed similar changes in the increase of Firmicutes and decrease in Bacteroidota at mid-point, a statistically significant increase in Firmicutes was only seen in the leaner twins who also had a significant increase in daily sugar consumption, especially at mid-point, possibly explaining the finding and highlighting the sensitivity to nutritional changes.

Unlike the three other major phyla, Campylobacterota increased from pre to post intervention. Studies on Campylobacterota in human microbiota are sparse. Higher relative abundances of Campylobacterota have been observed in SARS-CoV-2-infected people with severe respiratory symptoms [56] and Campylobacterota phyla have been associated with heme B synthesis pathways [57]. As the heme has a critical role in the ability of red blood cells to bind oxygen to be transported to organs and tissues, the increase in Campylobacterota may indicate better body oxygenation and aerobic capacity [58]. When twins were analyzed separately, the increase in Campylobacterota was only significant in the leaner twins, who also had higher aerobic capacity (VO_2peak_). Thus, the increase in Campylobacterota after the exercise intervention may reflect an increased demand and turnover of heme B.

Pooling together the results of our study, the transient changes at mid-point suggest that Firmicutes, Bacteroidota, Proteobacteria, and Campylobacterota are sensitive to changes in nutritional and exercise habits. This study strengthens the notion of an inverse relationship between alpha diversity and dietary fiber intake [59] and highlights the adaptability of microbiota at a phylum level to nutritional changes such as sugar consumption. The results of this study also further strengthen the proposition that Firmicutes are more effective in extracting energy and thrive in an environment of increased energy demands.

Although Firmicutes decreased from mid to post intervention, the relative abundance of *Lactobacillus*, *Megamonas*, and *CHKCL001* genus increased in the present study. Previously, it has been shown that elite athletes have a higher abundance of members of the Firmicutes phyla [60] and that aerobic exercise training with weight loss increases *Lactobacillus* [61]. Probiotics that contain members of the *Lactobacillus* genus have been shown to reduce upper respiratory tract infections [62] and improve endurance performance in healthy individuals without a history of professional athletic training [63].

*Megamonas* was strongly associated with exercise alone while both *CHKCL001* and *Lactobacillus* were associated with both exercise and nutritional variables such as sugar or its more specific sub-types, glucose and sucrose. *Lactobacillus*, however, was associated with exercise to a much greater degree than *CHKCL001*.

Our study shows that within the Firmicutes phylum, *Lactobacillus*, *Megamonas*, and *CHKCL001* are sensitive to exercise to a varying degree, but *Lactobacillus* and *Megamonas* also react to changes in nutritional habits, especially regarding sugar consumption. The effects occur even without a concomitant decrease in body weight. However, the effect of body weight may also be relevant to the response to exercise as only the leaner twins reached statistically significant increases in the relative abundance of *Lactobacillus* when analyzed separately.

*Helicobacter*, *Sellimonas*, *Enorma*, and *Rikenella* increased in both twin groups after training and showed a strong association with exercise alone. Previously, *Helicobacter*, *Sellimonas*, and *Rikenella* have been suggested to be related to positive changes in host health profile. *Helicobacter* is widely known for one of its subspecies, namely *Helicobacter pylori*, which is known for gastrointestinal infections [64]. However, in rat studies, fecal matter transplanted to non-exercising mice from exercising mice fed with either a high-fat diet or a normal diet caused increases in the abundance of *Helicobacter*, which indicates that exercise has a role in Helicobacter abundance [65]. *Sellimonas* have been thought to be a biomarker of the recovery of intestinal homeostasis [66], although increases in relative abundance have also been shown in chronic kidney disease patients [67]. The role of *Enorma* in health is unclear, but increased abundances of *Enorma* have been observed in morbidly obese patients with cholecystectomy, and lower abundances have been observed in patients with Blastocystis [68]. *Rikenella* has been shown to increase its relative abundance from an exercise intervention conducted on ApoE knockout mice with additional net positive health changes such as decreased obesity [69]. Our data indicate that *Helicobacter*, *Sellimonas*, *Enorma*, and *Rikenella* are malleable with exercise and may represent net positive changes in human health.

In our study, the relative abundance of *Cutibacterium* and *Staphylococcus* increased in both twins. *Cutibacterium* and *Staphylococcus* were similarly associated with nutrition and exercise. *Staphylococcus* has previously been shown to increase in prolonged physical stress [70]. In that study, however, the stress was a four-day-long ski-march. *Cutibacterium* is a commensal bacteria that is most commonly found in the skin [71]. The relationship between *Cutibacterium* and *Staphylococcus* and exercise is unclear as the studies linking these genera and exercise are sparse.

In our study, the twins differed significantly in body composition; all lived in Finland, and at baseline, the only significant nutritional difference between the twins, based on food diaries, was in dietary and insoluble fiber and soluble polysaccharide intake at baseline. There was no significant change in training adherence or duration from pre to mid and mid to post intervention or changes in body composition between the twin groups. These results would indicate that the effects of exercise are greatest at the genus level rather than at the phylum level. Overall, our findings further strengthen the notion that host gut microbiota is an ever-changing organism that responds to environmental changes independent of weight loss or genetics.

The limitations of this study were the small number of participants as monozygotic twins discordant for body weight are rare. No VO_2peak_ test was performed in the mid phase. The food diaries were completed by the participants and not by the researchers. When using food diaries, there is always the possibility of under-reporting. In our microbiota sensitivity analyses with additive calculation methods, not all assumptions of the model were fulfilled due to the much-skewed distributions due to several genera showing frequent zero abundance. Intestinal GU was calculated, standardized, and normalized to the unitary volume of the gut segment. Due to the challenging anatomy of the intestine, we could not measure whole intestine GU, which might give more information on the impact of overall intestine metabolism on glycemic control.

## 5. Conclusions

This study indicates that obesity impairs insulin-stimulated intestinal GU (intestinal uptake of circulating glucose from blood) independent of genetics. In the present study, long-term regular exercise training improved aerobic fitness and whole-body insulin sensitivity in both the leaner and the heavier co-twins. Though both twin groups exhibited some microbiota changes at the genus level, most changes in insulin-stimulated colon GU and microbiota composition were significant in the leaner twins.

## Figures and Tables

**Figure 1 nutrients-16-03554-f001:**
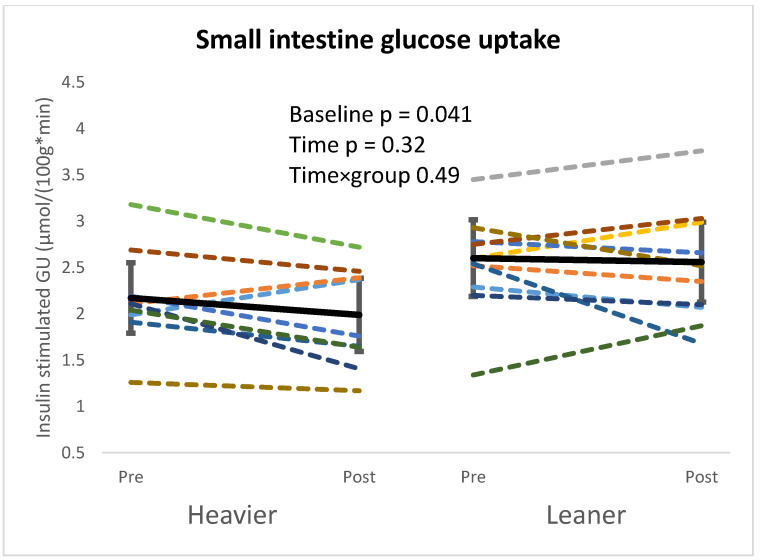
Heavier twins had lower small intestine GU at baseline, but no changes were seen after the intervention. Linear mixed model used for analysis. Heavier twins pre n = 11, post n = 9, and leaner twins pre n = 11, post n = 10. Twins from the same pair are colored with the same color.

**Figure 2 nutrients-16-03554-f002:**
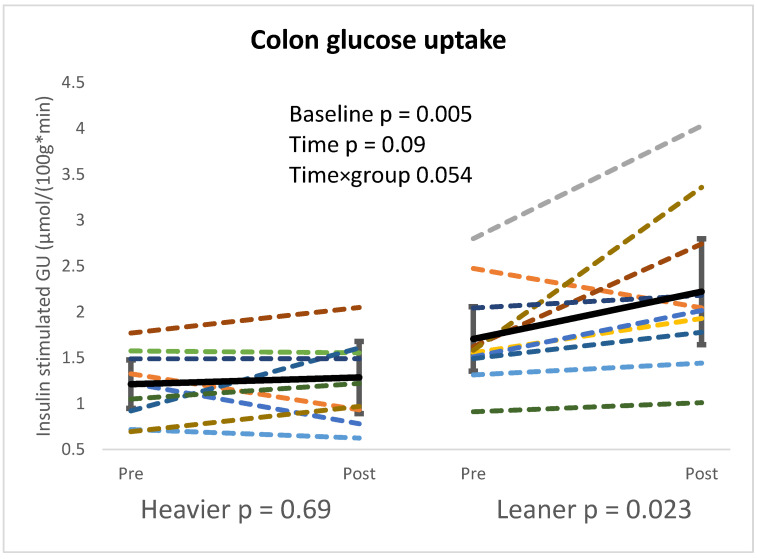
Higher colon glucose uptake (GU) was seen in heavier twins at baseline, and after the intervention, both twins saw an increase in colon GU, but leaner co-twins saw a greater increase. Linear mixed model used for analysis. Heavier twins pre n = 11, post n = 9, and leaner twins pre n = 11, post n = 10. Separate *p*-values represent the time effect when twin groups were analyzed separately. Twins from the same pair are colored with the same color.

**Figure 3 nutrients-16-03554-f003:**
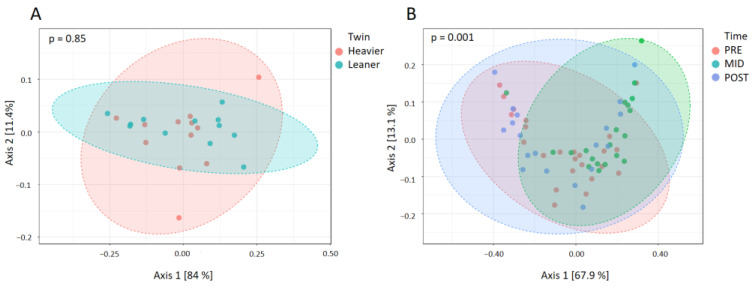
At baseline beta-diversity computed as based on Bray–Curtis index, Jensen–Shannon Divergence, and Jaccard index dissimilarity indices was no difference between the twins (**A**), but exercise training intervention caused a significant change (according to the three distance metrics) in beta diversity (**B**). Figures report PCoA plots based on Jaccard index at phylum level.

**Figure 4 nutrients-16-03554-f004:**
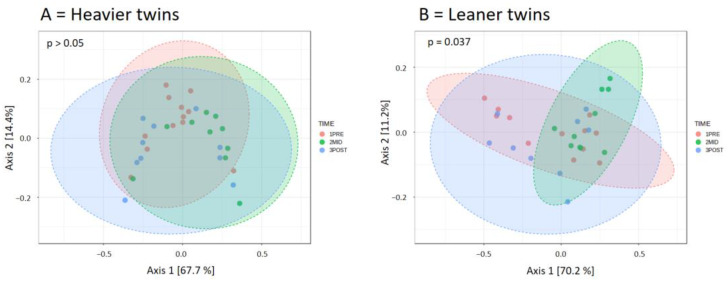
When the twin groups are analyzed separately, beta diversity computed based on Bray–Curtis index, Jensen–Shannon Divergence, and Jaccard index distance metrics over time changed significantly only in leaner twins. Figures report PCoA plots based on Jaccard index, at phylum level. (**A**) = heavier twins, (**B**) = leaner twins.

**Figure 5 nutrients-16-03554-f005:**
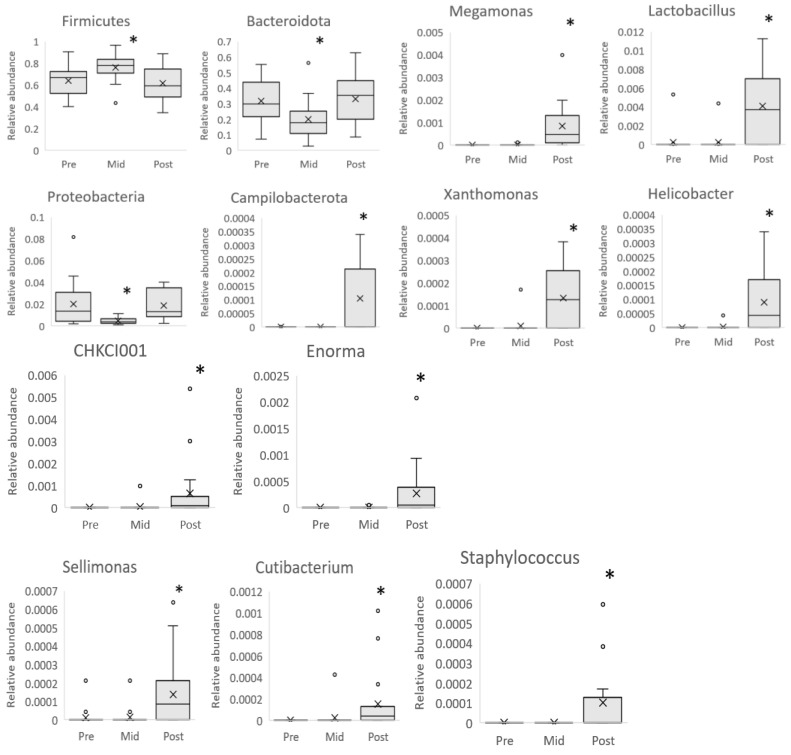
Bacteroidota and Proteobacteria levels decreased in the whole sample at mid-intervention but increased to baseline levels at the end of the intervention. The opposite change was observed for Firmicutes, where an increase was observed at mid-intervention, but levels fell back to baseline at the end of the intervention. After the intervention, there was an increase at the phylum level in *Campylobacterota* and at the genus level in *Megamonas*, *Helicobacter*, *Sellimonas*, *Lactobasillus*, *CHKCL001*, *Cutibacterium*, *Xanthomonas*, *Enorma,* and *Staphylococcus* compared to baseline in the whole sample. Linear mixed model used for analysis. * Statistically significant pFDR value (pFDR ≤ 0.05). Pre heavier n = 12, leaner n = 12. mid heavier n = 10, leaner n = 10. post heavier n = 10, leaner n = 9. The boxplots represent relative abundances. Box plot whiskers represent the maximum and minimum excluding outliers, dots represent outliers, X represents the mean of the data.

**Figure 6 nutrients-16-03554-f006:**
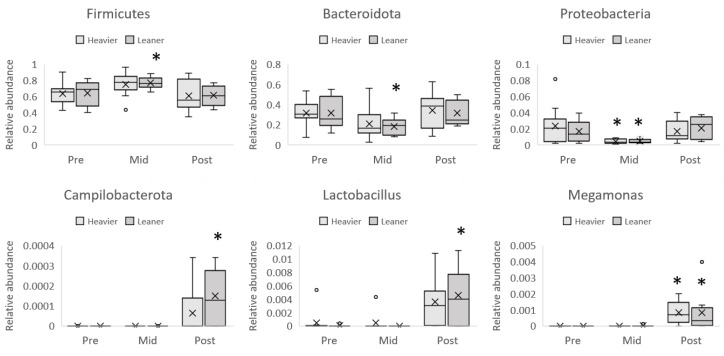
The transient (at mid-point only) increment of Firmicutes and reduction of Bacteroidota and Proteobacteria phyla were similar between twins but reached statistical significance in the leaner group only, with the exception of the reduction of Proteobacteria. Similarly, the increment of Campylobacterota at the end of the intervention was statistically significant in leaner but not in heavier twins. The increment of *Lactobacillus* genera reached statistical significance in the leaner twins only. Linear mixed model used for analysis. * Statistically significant pFDR value (pFDR ≤ 0.05). Pre heavier n = 12, leaner n = 12. mid heavier n = 10, leaner n = 10. post heavier n = 10, leaner n = 9. The boxplots represent relative abundances. Box plot whiskers represent the maximum and minimum, excluding outliers, dots represent outliers, X represents the mean of the data.

**Table 1 nutrients-16-03554-t001:** Subject characteristics of the leaner and the heavier twin groups before and after exercise intervention [mean (95% CI)].

	Heavier		Leaner		*p*-Value		
	Pre	Post	Pre	Post	Baseline	Time	Time × Group
n	12	11	12	10			
Male/female	4/8	4/7	4/8	4/6			
Age, years	40.4 (37.5; 43.4)		40.4 (37.5; 43.4)				
Weight, kg	108.7 (91.8; 125.7)	108.0 (93.1; 122.9)	86.4 (72.4; 100.3)	86.9 (72.6; 101.2)	0.001 *	0.95	0.37
BMI, kg/m^2^	36.7 (32.2; 41.1)	36.4 (32.4; 40.4)	29.1 (25.1; 33.1)	29.3 (25.3; 33.2)	0.001 *	0.92	0.41
Waist circumference, cm	117.7 (106.3; 129.2)	115.0 (106.8; 123.1)	96.5 (84.7; 108.3)	94.4 (82.3; 106.6)	0.001 *	0.17	0.74
Fat free mass, kg	35.9 (31.0; 40.7)	36.2 (32.1; 40.3)	33.1 (29.0; 37.2)	33.9 (30.6; 37.2)	0.003 *^†^	0.14	0.10
Fat mass, kg	45.5 (33.8; 57.3)	44.5 (34.6; 54.4)	27.8 (17.9; 37.7)	26.9 (14.4; 39.4)	0.001 *^†^	0.70 ^†^	0.97 ^†^
Visceral fat mass, kg	5.9 (4.5; 7.3)	5.5 (4.4; 6.5)	3.1 (2.0; 4.3)	3.2 (2.0; 4.4)	0.002 *^†^	0.07	0.29
Fat percentage, %	40.6 (35.5; 45.7)	40.0 (35.9; 44.1)	30.4 (24.0; 36.9)	29.5 (20.3; 38.7)	0.001 *	0.37	0.72
VO_2peak_, mL·kg^−1^·min^−1^	25.6 (22.7; 28.5)	28.3 (26.1; 30.6)	32.4 (27.3; 37.4)	35.1 (29.9; 40.2)	0.003 *	0.001 *	0.94
Triglycerides, mmol/L	1.4 (0.9; 1.9)	1.2 (0.9; 1.5)	0.8 (0.6; 1.0)	0.8 (0.6; 1.0)	0.040 *	0.54 ^†^	0.49 ^†^
Ffa, mmol/L	0.59 (0.51; 0.67)	0.68 (0.45; 0.91)	0.52 (0.34; 0.69)	0.54 (0.14; 0.94)	0.29	0.63	0.63
CRP, mg/L	2.8 (1.4; 4.2)	3.7 (1.3; 6.2)	2.1 (0.3; 3.9)	1.2 (−0.6; 3.1)	0.050 *	0.69 ^†^	0.50 ^†^
M-value, μmol/kg × min	23.1 (16.3; 30.0)	31.4 (20.4; 42.3)	37.6 (26.7; 48.5)	46.9 (31.7; 62.1)	0.007 *	0.022 *	0.82

*p*-value (linear mixed model) for baseline: within-pair difference before intervention, time: pre and post-difference in whole sample, time × group: did the training response differ within twin pairs. For M-value, small intestine GU and Colon GU heavier co-twins pre n = 11, post n = 9, and leaner co-twins pre n = 11, post n = 10. Abbreviations: BMI, body mass index; VO_2peak_, aerobic capacity; Ffa, free fatty acids; CRP, C-reactive protein; M-value, whole-body insulin sensitivity. * Statistically significant *p* value (*p* ≤ 0.05). ^†^ Logarithmic transformation.

**Table 2 nutrients-16-03554-t002:** Alpha diversity and average daily nutrient intake of the leaner and the heavier twin groups before, mid, and after exercise intervention [mean (95% CI)].

	Heavier			Leaner				*p*-Value		
	Pre	Mid	Post	Pre	Mid		Post	Baseline	Time	Time × Group
n	10	9	10	10	9		9			
Male/female	4/8	4/7	4/7	4/8	4/6		4/6			
Pielou Evenness	0.73 (0.70; 0.76)	0.73 (0.70; 0.76)	0.69 (0.65; 0.73)	0.75 (0.72; 0.78)	0.76 (0.74; 0.78)		0.72 (0.69; 0.75)	0.33	0.019 *	0.78
Chao 1	353.8 (312.3; 395.4)	350.0 (311.3; 388.6)	357.0 (301.7; 412.3)	339.1 (277.8; 400.4)	394.3 (358.3; 430.4)		414.3 (361.7; 466.9)	0.60	0.40	0.36
Dominance	0.04 (0.03; 0.06)	0.04 (0.03; 0.05)	0.05 (0.03; 0.08)	0.03 (0.02; 0.04)	0.03 (0.02; 0.03)		0.04 (0.02; 0.05)	0.25	0.14	1.00
Observed otus	347.9 (307.4; 388.4)	343.4 (306.4; 380.4)	345.0 (290.9; 399.1)	334.5 (273.2; 395.8)	387.1 (352.0; 422.3)		401.6 (349.3; 453.8)	0.63	0.53	0.37
Shannon	6.1 (5.8; 6.4)	6.1 (5.8; 6.4)	5.8 (5.3; 6.2)	6.2 (5.8; 6.7)	6.5 (6.3; 6.8)		6.2 (5.9; 6.6)	0.64	0.07	0.58
Simpson	1.0 (0.9; 1.0)	1.0 (1.0; 1.0)	1.0 (0.9; 1.0)	1.0 (1.0; 1.0)	1.0 (1.0; 1.0)		1.0 (1.0; 1.0)	0.31	0.23	0.83
Energy, kcal	2111.2 (1781.8; 2440.7)	2180.6 (1814.5; 2546.6)	2184.0 (1941.7; 2426.4)	1960.6 (1627.1; 2294.1)	2390.7 (1908.5; 2872.8)		2111.9 (1774.7; 2449.1)	0.29 ^†^	0.18	0.58
Fat, g	95.1 (74.2; 116)	92.1 (71.6; 112.6)	88.1 (70.3; 106.0)	87.7 (68.2; 107.2)	97.4 (73.5; 121.3)		93.8 (72.6; 115.1)	0.32	0.82	0.39
Carbohydrate, g	205.6 (161.8; 249.3)	212.6 (155.8; 269.4)	233.0 (193.8; 272.2)	179.69 (146.5; 212.9)	256.6 (196.8; 316.3)		209.8 (164.7; 255.0)	0.31	0.07 ^†^	0.28 ^†^
				Pre->Mid *p* = 0.030 ^†^*						
Protein, g	85.0 (71.48; 98.6)	95.3 (74.2; 116.3)	94.9 (80.7; 109.0)	92.3 (75.3; 109.2)	100.7 (79.7; 121.8)		92.9 (77.4; 108.4)	0.43 ^†^	0.38 ^†^	0.78 ^†^
Dietary fiber, g	19.3 (16.0; 22.5)	20.6 (15.4; 25.8)	22.9 (19.0; 26.9)	16.4 (12.4; 20.3)	20.5 (13.9; 27.0)		20.8 (15.7; 25.8)	0.053 *	0.054 ^†^	0.80 ^†^
Fatty acids, g	87.9 (68.1; 107.6)	86.1 (66.3; 105.9)	82.5 (65.3; 99.8)	80.9 (62.7; 99.2)	89.7 (66.4; 112.9)		86.7 (66.2; 107.3)	0.43 ^†^	0.89 ^†^	0.28 ^†^
Saturated fatty acid, g	34.7 (26.3; 43.0)	35.3 (24.4; 46.1)	32.9 (25.4; 46.1)	33.9 (24.5; 43.4)	36.7 (25.3; 48.1)		35.2 (27.2; 43.3)	0.70 ^†^	0.93 ^†^	0.59 ^†^
Cholesterol, g	280.1 (198.5; 361.6)	320.5 (144.6; 496.5)	255.5 (151.4; 359.6)	286.4 (165.4; 407.4)	358.2 (223.5; 492.9)		281 (205.0; 357.0)	0.72	0.25	0.50
Sterol, g	307.5 (248.3; 366.7)	290.8 (233.1; 348.5)	281.7 (225.9; 337.5)	252.8 (202.3; 303.2)	302.0 (235.5; 368.5)		311.3 (245.5; 377.1)	0.10 ^†^	0.69 ^†^	0.06 ^†^
Organic acids, g	4.0 (2.7; 5.3)	3.9 (2.6; 5.1)	4.5 (3.6; 5.4)	2.9 (2.0; 3.8)	4.2 (3.1; 5.4)		3.6 (2.7; 4.4)	0.06	0.55	0.07
Sugar, g	92.24 (67.6; 116.9)	90.9 (63.4; 118.4)	98.1 (70.4; 125.8)	69.1 (49.5; 88.6)	113.1 (84.7; 141.4)		95.1 (66.5; 123.7)	0.15	0.07	0.043
				Pre->Mid *p* = 0.004 ^†^*						
Fructose, g	10.7 (8.1; 13.4)	10.7 (7.9; 13.5)	14.2 (10.6; 17.8)	10.2 (3.8; 16.6)	12.1 (7.9; 16.3)		14.2 (8.6; 19.8)	0.34 ^†^	0.051 *^†^	0.50 ^†^
Glucose, g	12.4 (9.4; 15.5)	11.5 (8.1; 14.9)	15.4 (11.9; 18.9)	12.9 (6.5; 19.3)	16.4 (10.1; 22.7)		16.9 (9.7; 24.0)	0.89	0.024 *^†^	0.41 ^†^
Sucrose, g	53.8 (33.7; 74.0)	48.4 (22.7; 74.1)	49.6 (31.2; 67.9)	34.1 (23.2; 45.0)	65.4 (42.4; 88.3)		50.1 (33.4; 66.8)	0.13	0.16 ^†^	0.046 *^†^
				Pre->Mid *p* = 0.005		Mid->Post *p* = 0.016				
Soluble fiber, g	7.1 (5.9; 8.4)	8.9 (6.5; 11.2)	8.8 (6.8; 10.7)	6.7 (4.4; 9.0)	7.5 (4.2; 10.8)		8.1 (5.4; 10.9)	0.06 ^†^	0.18 ^†^	0.95 ^†^
Insoluble fiber, g	12.1 (9.6; 14.6)	11.5 (8.1; 14.9)	14.1 (7.7; 16.8)	9.7 (7.8; 11.6)	12.4 (8.0; 16.8)		12.6 (9.8; 15.6)	0.034 *^†^	0.09 ^†^	0.48 ^†^
Soluble polysaccharide, g	4.5 (3.6; 5.4)	3.7 (2.7; 4.6)	4.9 (3.9; 5.9)	3.7 (2.5; 4.9)	4.2 (3.2; 5.2)		4.4 (3.2; 5.5)	0.030 *^†^	0.16 ^†^	0.16 ^†^
Salt, g	7.5 (5.8; 9.3)	7.9 (6.3; 9.5)	8.7 (7.4; 10.1)	7.0 (5.6; 8.3)	9.9 (5.8; 14.0)		8.1 (4.5; 11.8)	0.62 ^†^	0.07 ^†^	0.78 ^†^

*p*-value (linear mixed model) for baseline: within-pair difference before intervention, time: pre, mid, and post difference in whole sample, time × group: did the training response differ within twin pairs. Heavier co-twins pre n = 12, mid n = 10, post n = 10, leaner co-twins pre n = 12, mid n = 10, post n = 9. * Statistically significant *p* value (*p* ≤ 0.05). ^†^ Logarithmic transformation.

## Data Availability

Datasets from the current study are available from the corresponding author upon reasonable request.

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
