# Peer review of "Regular Exercise Training Induces More Changes on Intestinal Glucose Uptake from Blood and Microbiota Composition in Leaner Compared to Heavier Individuals in Monozygotic Twins Discordant for BMI"

_nutrients, 2024, doi:10.3390/nu16203554_

Round 1
Reviewer 1 Report
Comments and Suggestions for Authors
This very interesting study involving glucose metabolism in monozygotic twins discordant for obesity provided a rare opportunity for authors to evaluate effect of obesity and exercise on transport of glucose from the blood into abdominal organ systems. There are also a lot of descriptive data on gut microbiota that are at best correlative with their physiological observations.
General comments:
1. There is a misunderstanding of glucose transport terminology that needs clarification. The human disease glucose-galactose malabsorption arises from the absence of the transporter SGLT1 from the apical membrane of small intestinal cells, causing massive diarrhea as there is no glucose uptake. The colon is unable to absorb (uptake) glucose because it does not express SGLT1 (Lostao MP, Wright EM et al, Function (Oxf). 2021). In normal conditions, glucose is transported from the diet via SGLT1 into cytosol, and from cytosol via GLUT2 into the blood. GLUT2 in the apical membrane has largely been disproven (see Ann Rev Nutr reviews). Intestinal glucose uptake, by widely accepted convention (see reviews in Physiology journals on intestinal glucose uptake), means glucose absorption from the intestinal lumen (thru the apical membrane of the intestinal epithelial cell then traverse cytosol then cross the basolateral membrane) into the blood and eventually into the portal vein. It does not refer to glucose transport from systemic circulation into organ systems. Moreover, the authors here use FDG as tracer injected into the antecubital vein (Heiskanen et al 2021) when blood glucose is stable at 5 mM and FDG is a GLUT2-specific PET tracer, so their method does determine movement of blood glucose into GLUT-2 expressing abdominal organs, not glucose uptake from the lumen. This should be made clear in the manuscript.
2. The major technical issue is that GLUT2 is extensively expressed not only in the intestine, but also in the liver, pancreas and kidneys. The authors did not show samples of their scans, so it is not clear “what region of interest” represent, because the image intensity is affected by liver, pancreas and renal FDG uptakes which may also be prominent in the abdominal regions. How did the authors correct for this “background, non-intestinal, non-colon FDG”?
3. There are numerous studies dating back to early 1900’s that consistently find obese individuals to have significantly longer intestines that nonspecifically increase nutrient absorption from the lumen into the cell. How did the authors account for this anatomical disparity that would impact amount of FDG transport from blood into small intestine and colon?
4. Nonpancreatic GLUT2 is a facilitative transporter and is not insulin-sensitive. Its main function in the small intestine is to move glucose out from the cytosol into the blood, if cytosolic glucose (as what happens during a meal) is > 5 mM. Its main function in the other organs is to take up glucose from the blood to feed cells. Cytosolic glucose in the small intestinal enterocytes between meals will be low, and at 5 mM blood glucose, there is GLUT2-mediated FDG transport from blood into enterocyte as GLUT2 allows the glucose concentration to reach balance on the bilateral sides of the cellular membrane. Thus, the “uptake” or movement of glucose into the small intestinal enterocyte depends mostly on time after meals which impact glucose concentration in the cytosol, while the movement of glucose into colonocytes (which cannot “feed” itself via SGLT1) is independent of time after meals. This should be accounted for in interpreting the data.
5. The changes in microbiota are correlated with changes in systemic glucose homeostasis, but do not cause them. There are no data to indicate that a change in microbiota will cause favorable outcomes. Thus, the phenomenological conclusion that favorable changes in insulin-stimulated colon glucose uptake and microbiota composition were significant in the leaner twins is speculative.
6. The conclusions that long-term regular exercise training improved aerobic fitness and whole-body insulin sensitivity in both the leaner and the heavier co-twins and that both twin groups exhibited some microbiota changes at the genus level are supported by the data. The conclusion that obesity impairs insulin-stimulated intestinal GU independent of genetics should be revised in light of comments 1-4.
7. This manuscript is well written.
Author Response
We Thank the reviewer for taking the time to review this manuscript. Please find the detailed responses in the word file attached.

Reviewer 2 Report
Comments and Suggestions for Authors
Genetic, diet and environmental factors may influence the composition of intestinal microbiota. In addition, recent research has shown that exercise can increase the number of beneficial microbial species. As a result, this manuscript explores a significant area of study regarding the microbiota composition in ten pairs of monozygotic twins and its relationship with several factors, including intestinal glucose uptake and regular exercise training.
To improve the quality of the manuscript, I have some suggestions for the authors:
- The correspondence author's name and email address are not easily visible in the authors list.
- The authors should add more data regarding the relationship between genetics and gut microbiota in lines 80-85.
- Kindly reorganize the abstract of the manuscript. It should be shorter (no more than 250 words - please see the Nutrients journal recommendations).
- To avoid excessive lines, kindly decrease the size of letters/the size of the numbers in tables
- Line 393 – kindly add supplementary references (the authors refer to more than one study, but one reference is listed).
- firmicutes should be replaced with Firmicutes in line 448
- Kindly revise the English language. Some sentences are heavy to read (please see lines 39-42, 70-73 or 420-422).
- Please write the reference list according to the recommendations of the Nutrients journal. https://www.mdpi.com/journal/nutrients/instructions
Comments on the Quality of English Language
Kindly revise the English language. Some sentences are too heavy to read.
- ’’ Exercise has been shown to alter gut microbiome composition[10] and markers of healthy gut microbiome, such as the diversity of the host gut microbiome, which was reported to be greater in athletes compared to sedentary controls in some[11, 12], but not all studies[13].’’
In this study, the microbiota data was derived from fecal samples and in pigs fecal microbiota has been shown to differ significantly from intestinal microbiota from various parts of the gastrointestinal tract. (lines 420-422)
Author Response

(The authors gave the same response as above.)

Round 2
Reviewer 1 Report
Comments and Suggestions for Authors
The authors have mostly responded to all my concerns. This article offers a new perspective on intestinal glucose metabolism by providing interesting information on glucose transport from the blood. My remaining suggestion is to also include "uptake from blood" in the title, not only in abstract, because it actually highlights the novelty of their work. Based on the current title, readers interested in (luminal) gut microbiota will immediately assume the authors refer to glucose uptake from the lumen. A PubMed search in the past 70 years using "intestinal glucose uptake" retrieves thousands of studies pertaining almost entirely to absorption from the food side. This reviewer has also been asked by several journals to comment on several articles on exercise, microbiota and glucose absorption, and most of these manuscripts refer to uptake from lumen; those that did not were clear about the systemic source of glucose.
Author Response
We Thank the reviewer for taking the time to review this manuscript. Please find the new responses in the word file attached.
